# A Novel Diagnostic Framework with an Optimized Ensemble of Vision Transformers and Convolutional Neural Networks for Enhanced Alzheimer’s Disease Detection in Medical Imaging

**DOI:** 10.3390/diagnostics15060789

**Published:** 2025-03-20

**Authors:** Joy Chakra Bortty, Gouri Shankar Chakraborty, Inshad Rahman Noman, Salil Batra, Joy Das, Kanchon Kumar Bishnu, Md Tanvir Rahman Tarafder, Araf Islam

**Affiliations:** 1Department of Computer Science, Westcliff University, 17877 Von Karman Ave 4th Floor, Irvine, CA 92614, USA; chakraborttyjoy9@gmail.com (J.C.B.); a.islam.585@westcliff.edu (A.I.); 2Department of Computer Science and Engineering, Lovely Professional University, Phagwara 144411, Punjab, India; salilbatra40@gmail.com; 3Department of Computer Science, California State University, 5151 State University Dr, Los Angeles, CA 90032, USA; inoman@calstatela.edu (I.R.N.); kbishnu@calstatela.edu (K.K.B.); 4Department of Pharmacology, School of Pharmaceutical Sciences, Lovely Professional University, Phagwara 144411, Punjab, India; joydas5852@gmail.com; 5College of Engineering and Technology, Westcliff University, Irvine, CA 92614, USA; tanviraditto90@gmail.com

**Keywords:** vision transformer, deep learning, CNN, VGG19, IoMT and classification

## Abstract

**Background/Objectives:** Alzheimer’s disease (AD) is a progressive, neurodegenerative disorder, which causes memory loss and loss of cognitive functioning, along with behavioral changes. Early detection is important to delay disease progression, timely intervention and to increase patients’ and caregivers’ quality of life (QoL). One of the major and primary challenges for preventing any disease is to identify the disease at the initial stage through a quick and reliable detection process. Different researchers across the world are still working relentlessly, coming up with significant solutions. Artificial intelligence-based solutions are putting great importance on identifying the disease efficiently, where deep learning with medical imaging is highly being utilized to develop disease detection frameworks. In this work, a novel and optimized detection framework has been proposed that comes with remarkable performance that can classify the level of Alzheimer’s accurately and efficiently. **Methods:** A powerful vision transformer model (ViT-B16) with three efficient Convolutional Neural Network (CNN) models (VGG19, ResNet152V2, and EfficientNetV2B3) has been trained with a benchmark dataset, ‘OASIS’, that comes with a high volume of brain Magnetic Resonance Images (MRI). **Results:** A weighted average ensemble technique with a Grasshopper optimization algorithm has been designed and utilized to ensure maximum performance with high accuracy of 97.31%, precision of 97.32, recall of 97.35, and F1 score of 0.97. **Conclusions:** The work has been compared with other existing state-of-the-art techniques, where it comes with high efficiency, sensitivity, and reliability. The framework can be utilized in IoMT infrastructure where one can access smart and remote diagnosis services.

## 1. Introduction

Alzheimer’s disease (AD) is a complex neurodegenerative disorder that significantly affects memory, cognition, and behavior, primarily in older adults [1,2]. It is the leading cause of dementia, accounting for 60–80% of dementia cases globally, including a significant proportion in the United States [3]. They are characterized by the presence of extracellular amyloid deposits, which are amyloid beta (Aβ), and intracellular neurofibrillary tangles, which are tau protein. In addition to that, AD leads to the death of the brain cells and thus a progressive decline in cognitive function [4,5]. Disease symptoms include forgetfulness and difficulty with tasks; later, patients may get lost, have problems with speech and language, become emotionally unbalanced, and lose capacity for self-care [6]. As shown in Figure 1, early-stage symptoms of AD include memory loss, confusion with time and location, difficulty solving problems, withdrawal from social activities, poor judgments, and changes in vision. Additionally, it explains the main risk elements that contribute to the progressive disease, i.e., age, diabetes, genetic predisposition, obesity, smoking, and brain injuries.

In AD, the level of brain damage increases across various degrees of cognitive impairment during the disease course [7]. As for normal cognitive (NC), the brain is functioning well, but certain alterations might start a long time before the first sign appears [8]. Research shows that patients with NC do not lose as much as 1 percent of their brain volume per year [9]. Cases with Mild Cognitive Impairment (MCI) have a faster rate of shrinkage of the brain, which causes them to lose between 1 and 3% of their brain mass annually, commonly in the hippocampal region responsible for memory [10].

As soon as a person enters AD, brain atrophy becomes much more rapid, the losses reaching from 3 to 5% per year, which indicates active neuronal degeneration [11]. Due to its distinctive neuropathologic profile, the damage in AD especially occurs in locations that control memory, language, and other executive abilities, which are thus characteristically impaired in this disease [12]. Figure 2 shows a comparison of the normal brain and the brain affected with AD blood brain barrier (BBB) disruption and neuroinflammation are central to the pathogenesis of AD and are associated with impaired clearances of Aβ peptides. The end result is the appearance of Aβ plaques and neurofibrillary tangles that induce massive neuronal damage. These changes become pathological and over time result in large volumes of brain shrinkage, such as in the hippocampus (which governs memory and functions of the mind). It further shows a deep learning (DL)-based model for AD detection and classification. The computational approach presented here draws synergies from advanced computational techniques to analyze patterns seen in affected brains for accurate detection and reliable performance validation. The potential for better diagnostics and therapeutics arises from the combination of pathophysiological insights and artificial intelligence (AI)-based detection methods.

In the U.S.A., AD is a major public health problem, with 6 million Americans estimated to live with the disease and expected to increase to nearly 14 million by 2060 [13]. While traditional AD diagnosis, with clinical assessment, history, and neuroimaging, is often inaccurate and too late in diagnosis. Nowadays, approximately two-thirds of seniors are care-receiving ones, and approximately one-third of them die with AD or another type of dementia, which is not surprising at all, given the necessity in terms of the development of better diagnostic and therapeutic means. Most cases of AD occur in people who are over the age of 65, albeit the primary risk factor continues to be age [14,15]. This poses an immense problem for the healthcare providers who would like to administer the therapy before the disease starts developing irretrievable stages [16]. In the U.S., approximately 12–18% of those aged 60 and older are living with MCI, with early detection of MCI being important so that we can intervene before AD becomes full blown [17,18].

Additionally, using AI models, medical professionals can detect vulnerable patients, in particular during the MCI stage, when intervention could stop or deter subsequent cognitive decline [19,20]. In the last few years, Machine Learning (ML) and especially DL have made recent advances in the early detection and diagnosis of AD [21]. We have shown that ML techniques have the potential to analyze large datasets and identify patterns that can be difficult to observe by clinicians [22,23]. Newer-generation technologies are being used to analyze many kinds of data, including genetic information, imaging data, and cognitive scores, in order to accurately predict risk and initiate earlier interventions [24]. Furthermore, deep learning models that have exhibited the capability of processing complex datasets like MRI, PET scans, and even other more complex neuroimaging data have substantially improved AD detection accuracy [25]. Despite these advances, challenges remain, particularly in making these models interpretable and explainable to clinicians, as the “black box” nature of many ML models hinders their widespread adoption in real-world clinical settings [26]. In the U.S., numerous research initiatives and studies are focused on using ML and DL models to improve AD detection [27]. Such models can thus discriminate between different cognitive states, for instance, NC, MCI, and AD [28]. There are researchers working on creating explainable artificial intelligence (AI) so these models can not only predict outcomes but also explain why these predictions are so, so that these models can effectively explain the factors leading to them [29]. For instance, we have seen recent studies employ approaches such as SHapley Additive exPlanations (SHAP) to provide both global and local explanations of the predictions made by the models to help the clinician know which features most influenced the decisions [30,31]. However, additional work will be needed to enhance the transparency, interpretability, and clinical applicability of these models.

DL techniques have been explored in this work to classify AD, leveraging a review of the recent works, challenges, and a growing literature in the USA [32]. It will also evaluate the status of current approaches and the future promise of AI-driven diagnostic tools in the application to AD patient management in the clinical setting [33]. By using large datasets, multimodal information, and interpretable AI ways, researchers want to get more accurate, more reliable models to predict the progression of AD and provide early intervention and better outcomes [34]. CNN models are very efficient and are widely being used for image classification-based tasks where various CNN-based diagnosis frameworks have been developed and applied in medical imaging. Although CNN is good at classification tasks, ViT or vision transformer can handle the task more efficiently, where it has more advantages over CNN [13]. ViT uses patches and a self-attention mechanism that allows it to have a long-range relationship. ViT has good scalability with less inductive bias. CNN extracts features from input images in a hierarchical manner where ViT follows global feature learning. Although CNN is widely popular and is highly efficient for image classification applications, ViT comes with more advantages when the dataset is large with a huge volume of instances [31,32]. Ensemble technique helps combine the performance of each model and offers better performance than using the models individually. In a weighted average ensemble, weight is assigned based on the individual performance that can use the combined potential of each model performance-wise and comes with effective results [16].

In this work, the following have been performed:(1)An extensive literature study has been performed where related works have been thoroughly studied to know about the existing state-of-the-art techniques and to identify current research gaps.(2)A classification framework has been designed based on vision transformer (ViT-B16) and three fine-tuned CNN models (VGG19, ResNet152V2, and EfficientNetV2B3) with modified dense layer architecture trained with almost 0.234 million brain MRI images.(3)A weighted ensemble approach with Grasshopper optimization algorithm applied on developed vision transformer and CNN models to build an efficient, robust, and remarkable framework for multiclass classification of Alzheimer’s level.(4)An innovative model of AI integrated IoMT architecture has been designed and proposed where the designed classification framework applied in cloud server can play a significant role in telemedicine where one can access smart diagnosis service remotely.(5)A comparative analysis with proper facts and figures has been performed with the existing state-of-the-art work that ensures the strengths and novelty of the framework in terms of architecture design and performance.

In Section 2, a literature review is conducted, where recent related work with their key findings and limitations are presented and properly explained. Section 3 deals with the methodology, where design and development of the proposed framework are presented. Section 4 shows the results of the experimental analysis and discussion regarding the facts and figures regarding the performance of the proposed work. Lastly, a conclusion is drawn in Section 5, where a summary of the proposed work is presented with future scope.

## 2. Related Work

Innovative approaches and technologies that have been adopted for the identification of AD have been illustrated in a recent study. Yogesh et al. [19] analyzed the effectiveness of using ML algorithms in KNN and SVM for diagnosing mental disorders by assessing the output generated by MRI images; moreover, they found that KNN is more accurate than SVM in this process. Such technologies are vital for diagnosis in the early stage [19]. Altogether, the current study contributed to an affordable merged platform that incorporates the cognitive assessments with the noninvasive physiological monitoring to document the advancement of AD by Panda et al. [20]. As Qu et al. [21] suggested, modifying the immunostaining process to a univariate neurodegeneration biomarkers approach that enhances early identification and using it in the GCN model that authors identified provided high percent classification between cognitively impaired and nonimpaired partakers in the study [21]. Tushar et al. [22] proposed a novel hybrid model combining both logistic regression and decision trees that has increased the prediction value greatly, employing data from the OASIS database. The utility of DL techniques was tested by Alshammari et al. [23], who used a CNN model for accurate differentiation between the stages of AD from the MRI data with a high classification rate. Thatere et al. [24] pointed out the important aspects of biomarkers and interventions in ML strategies for AD diagnosis and highlighted data deficits. In another study, Sevcik et al. (2022) [25] were able to note through a systematic review of conversational datasets for AD prediction that increased data quality outweighs increased quantity for predicting the condition using natural language processing. The two related studies of Gurrala et al. [26] and Kumar et al. [35] pointed out that the conventional architecture of CNN, which facilitates MRI data handling, offers extremely high classification rates. Zubair et al. [36] rightly highlighted the role of the early detection strategies; Saxena et al. [37] narrated the recurrent DL progress while also noting the current existing challenges with practitioners. Classification of neuroimaging datasets was the subject of Archana B et al. [38], who recorded an impressive accuracy that serves as an indication that intervention should be promptly initiated. Liu [39] enhanced and expanded the classification by integrating the hippocampal MRI with the whole-brain MRI using an attention-enhanced DenseNet model. Han et al. [40] proposed an optimization in the feature extraction methods using LightGBM because of the high and better classification accuracies noticed in real datasets. Functional MRI is also used, and in a recent study, Maitha Alarjani [41] used a patch-based approach, in which functional MRI is correct; thus, there is a need to find key features for efficient diagnosis urgently. Sanjeev Kumar et al. [42] listed the CNN architectures to outperform the others in stages of AD classification, and Mishra et al. [43] described the early prediction problems of the different datasets for the ML approach. Lu et al. [44] introduced a multiclassification framework to enhance the measurement accuracy, which would in turn minimize the imaging overlap between AD and MCI using the ConvNeXt network. In the work of Prajapati et al. [45], deep neural solutions were identified for binary classification with deep neural networks to achieve over-average validation accuracy rates compared to other methods. Shi et al. [46] attempted to use a generative adversarial network framework to enhance diagnosis for tau PET images, noting that there is individual region analysis that needs to be performed. A very comprehensive overview of the role of AI in qualitative improvement for AD diagnosis using medical imaging was presented by Arulprakash A et al. [29]. According to Neetha et al. [47], the Borderline-DEMNET framework is suggested, and a good degree of accuracy is evaluated where stages of AD are concerned. Chander Prabha [48] also laid emphasis on early detection following an optimal MRI scanning method while achieving the desired % classification. MRI plays an important role in the detection of structural abnormalities, remarked Ruchika Das [49], with hippocampal segmentation providing better patient classification. Yuan et al. [50] compared different conventional ML methods and successfully classified them based on cortical features; Navarro et al. [51] established daily living activity tasks as possible screening tools for the assessment of cognitive impairments in the AD population. In the perspective of continued advancement of identification procedures, Alatrany et al. [52] and Bhargavi et al. [32] also emphasized the important contribution of numerous ML algorithms on enhancing the diagnostic efficiency as well as monitoring of AD stages. Wang et al. [53] attempted to discuss the features of DL in AD detection by using sMRI. In this review, the methodologies proposed were categorized based on input types, and issues regarding the difficulty of spatial modeling and variability of performance were also discussed while addressing future recommendations for further DL diagnostic models. The study by Kayalvizhi et al. [54] has identified that while using the mass VGG16 CNN model, the possibilities of accurate diagnosis of AD from MRI reach 96.75% for the DL model, which means it is applicable in the neuroimaging analysis. According to Akter et al. [55], different types of dementias were identified using the Extreme Gradient Boosting model, with an accuracy of 81% for the feature “ageAtEntry’“’ as the biomarker for the first diagnosis of dementia. In the study of Pallawi et al. [56], extreme emphasis was placed on the early diagnosis using a framework using transfer learning, and combining it with the EfficientNetB0 model achieved 95.78% early efficiency in classifying four stages of AD that achieved general milestones in the dataset for DL applications. Islam et al. [57] studied MRI’s contribution to AD diagnosis through a focus on the hippocampus and put forward YOLO-based models for automated hippo detection whereby positive outcomes have been achieved, helping to advance early recognition. Zhou et al. [58] designed a Chinese game application that combines cognitive tests, which may reflect patterns of cognitive decline useful for early AD identification. Rauniyar et al. [59] used NLP to review the rhetorical fluency in AD patients to set up a referential platform for further comparisons. Feed-forward neural networks have been analyzed by Patra et al. [60] on the OASIS dataset to make assessments that feed-forward neural nets are capable of predicting AD better. The various ML algorithms were reviewed and compared by Varma et al. [61] for their study’s purpose of patient data classification to achieve maximum predictive accuracy in identifying the presence of AD. MCI was split into three subclasses by Jiang et al. [62] using KNN accompanied with an enriched LSTM neural network for higher accurate disease prognosis. Bushra et al. [63] proposed a feature-level fusion approach for which two CNN architectures were used and received accuracies of 94.39% for MCI and 97.90% for AD as compared to the normal control group and were higher than current methods. Subha R et al. [64] have also stressed the need to bring about more efficient early AD diagnosis through the use of a novel hybrid ML model with higher accuracy achieved through the integration of particle swarm optimization. More recently, Buyrukoglu [65] underlined the need for early detection, demonstrating that this is possible with the help of ensemble learning methods, the best of which reached a 92.7% classification rate using the AdaBoost ensemble. Tripathy et al. [33] proposed a new architecture of multilayer feature fusion-based deep CNN for better prediction and found better accuracy of 95.16% with multiple scale features. Chandra et al. [66] tested the Naïve Bayes classifier using the brain volume percentages as extracted biomarkers and reported reasonably good classification performance. A new CNN model for the diagnosis of AD has been developed recently by Talha et al. [67] with sufficient accuracy and F1 score for the early tracking. Bharath et al. [68] have categorized and analyzed the various ML modeling techniques, where they expounded that the utilized SVM model was helpful in early AD and was accompanied with a high accuracy ratio of 98%. According to Sindhu et al. [69], neural network methodologies for AD detection need elaboration, and automated methodologies that improve the degree of precision should be advocated. Jansi et al. [70] analyzed some DL models, InceptionV3, which has achieved an 87.69% accuracy in their maximization of dataset sets. Yin et al. [34] introduced the novel SMIL-DeiT network for classifying stages of AD, reaching an accuracy rate of 93.2%, which is superior to the existing approaches. In Sushmitha et al. [71], a genetic algorithm with multi-instance learning has been used for separate 3D MRI feature extraction and overfitting problems. Kadyan et al. [72] tested speech signal patterns for the early identification of AD and recorded an accuracy of 98.3% using a random forest classifier. In order to understand and overcome the deficiencies of the existing studies about stage-specific brain pattern analysis, Peng et al. [73] put forward the concept of the SF-GCL model. According to STCNN proposed by Anjali et al. [74], the model showed shorter average execution time, superior accuracy, and F1 scores in the identification of AD. Yu et al. [75] have put forward a method known as a supervised deep tree model (SDTree) to solve this problem because current diagnostic technology cannot capture the continually progressive nature of AD. Specifically, by employing the nonlinear reversed graph embedding, their proposed approach deems the progression of AD as a latent tree structure model to enhance the prediction and achieve favorable performances on the ADNI dataset. Finally, Jin et al. [76] described an unsupervised DL model using adversarial autoencoders to identify brain atrophy with 94% accuracy as a classifier. Soni et al. [77] proposed a verb fluency task for the detection of AD with 76% accuracy using random forest, whereas Suttapakti et al. [78] improved feature extraction for AD classification using two-dimensional variational mode decomposition with 94.44% accuracy. Basher et al. [79] used the volumetric features from hippocampi, going up to 94.82% accuracy, and added more to the development of the diagnostic tools for AD. Altogether, the results of recent research investigations signal the paramount importance of early and correct identification, which is the most important for intervention approaches. Further, as the models are improved and new methodologies are devised by researchers, the potential to increase the diagnostic accuracy and ultimately, the benefits of diagnosis to the points of care, increases dramatically. Thus, AD detection now and in the future is laying great emphasis on the integration of the most advanced technologies, such as ML and DL. Table 1 shows a summarized representation of the existing related works.

## 3. Proposed Methodology

In this section, a brief representation has been given regarding the proposed framework, including dataset information, preprocessing, model architectures, design, and development of a classification framework with functional working methods, etc.

### 3.1. Dataset Description

To develop deep learning models, a huge amount of data is required. To ensure consistent performance with high sensitivity and reliability, data should be collected from a renowned and trusted source. To train models for developing the proposed framework, ‘OASIS Alzheimer’s Detection’ dataset was used. The collected dataset is considered an authentic and benchmark dataset that has been collected from Kaggle, which is licensed under Apache 2.0. The dataset includes 87,390 MRI images consisting of four classes: Mild dementia (5002 images), Moderate dementia (488 images), Nondementia (67,200 images), and very mild dementia (13,700 images). Figure 3 shows a sample image from the no dementia class from the same dataset of 24 × 24 size.

### 3.2. Preprocessing

To enhance the performance of the model through reducing data complexity, preprocessing plays a vital role. Data needs to be preprocessed before being fed to models. Preprocessed data ensures promising performance, removing issues like overfitting. The input images from the dataset are of 176 × 208 size, which must be resized before fitting. Moreover, the DL model works well with a large amount of data where augmentation techniques can play vital roles.

Figure 4 shows preprocessed data samples having the same size for all classes. Augmentation technique helps to produce two types of data—synthetic data and augmented data. Synthetic data is a type of artificially created data that has been created through generating algorithms. Augmented data are generated by modifying the existing data. Synthetic data is required to be generated when original data instances are very few in the original dataset. In the selected dataset, almost 87,390 images are already there for which synthetic generation is not required. Augmented data has been generated after applying several image transformation techniques like rotating, zooming, flipping, shearing, shifting, cropping images, etc. The third class has already 67,200 images; that is why augmentation was not performed for that class. Class 4 (Very Mild Dementia) has 13,700 images, which have been increased five times to get 68,200 images. Class 2 (Moderate Dementia) has only 488 images that must generate a class imbalance issue.

Class 4 has been scaled up to 20 times and gets 48,800 images. Class 1 (Mild Dementia) has been increased 10 times and comes with 50,020 images. The comparison has been presented in Figure 5 to indicate the difference between before and after augmentation. Before applying the data augmentation technique, there are a total of 87,390 images, and after applying the augmentation technique, the number becomes 234,220, which is 2.6 times more than the amount of the original dataset offers.

### 3.3. CNN-Based Classification

Convolutional neural networks are widely used in image classification applications for rich and remarkable performance. CNN architecture is basically divided into two parts—feature extraction and classification. The feature extraction part is responsible for extracting the required features from the input images. The classification part classifies the images based on the received features. Figure 6 shows the basic architecture of CNN, where convolutional and pooling layers are there for the feature extraction part, and the dense layer is responsible for the classification part.

Table 2 shows a comparative analysis of three CNN models- VGG19, ResNet152V2, and EfficientNetV2B3, based on various parameters [35].

To develop the proposed farmwork, besides using a vit-b3 model, three CNN models have been developed, which are VGG19, ResNet152V2, and EfficientNetV2B3.

VGG19, ResNet152V2, and EfficientNetV2B3 have been selected for this study because each model brings unique strengths to medical image classification. VGG19 is known for its deep and straightforward architecture that makes feature extraction effective. ResNet152V2, having its residual connections, overcomes issues like vanishing gradients and assists better training in deep networks. EfficientNetV2B3 offers efficient scaling, maintaining a balance between model speed, size, and accuracy. These three models were found ideal, whereby combining the three models with their complementary strengths—efficiency, depth, and performance—a well-rounded and robust framework can be developed for medical image classification.

VGG19 shows more promising performance than the other two CNN models. Table 3 shows the shape and number of parameters where the feature extraction part was taken from the predefined VGG19 architecture, which is mentioned as functional. The dense layer of the pretrained model was modified according to the desired output requirements, which is shown in Figure 7.

### 3.4. Architecture of Vision Transformer

Transformers are deep learning models that are designed for handling sequence-to-sequence tasks, being widely and effectively used in natural language processing (NLP) and computer vision. To analyze the relationships between elements in a sequence, transformers use a self-attention technique. Unlike traditional recurrent models, transformers can perform parallel processing to improve efficiency. Application in text generation, language translation, and image recognition, different prominent models like BERT, GPT, and Vision Transformer (ViT) are being used, which were built based on transformers. Their scalability and effectiveness have revolutionized AI applications across multiple domains. Figure 8 shows the basic architecture of the transformer, which has two parts—the upper one is the encoder and the lower one is the decoder. The input data is transformed into vector representations for each word through an embedding layer.

Since transformers do not process sequences in order, positional encodings are added to these embeddings to preserve the word order. The core of the attention mechanism relies on three learnable vectors: Query, Key, and Value, where a query is compared with a key to retrieve a corresponding value. The dot product of Query and Key creates a score matrix, showing how much attention each word should give to others, with higher scores indicating more attention. The scores are scaled to prevent large values that could destabilize gradients. Next, the scores are passed through softmax, turning them into probabilities that highlight the most important words. These probabilities are then multiplied by the value vectors, allowing the keywords to have a stronger influence. The resulting vectors are concatenated and passed through a linear layer for further processing. Since attention is calculated independently for each word, multiple parallel attention heads are used to capture different relationships in the data. Outputs are added to residual connections (to aid gradient flow) and normalized with LayerNorm to stabilize the learning process.

The output then passes through a feed-forward network for a richer representation. Another round of residual connections and LayerNorm ensures the output remains stable. This completes the encoder, which creates a full representation of the input sequence. The decoder uses the encoder’s output and previous time-step inputs to generate the final sequence, applying masked multiheaded attention to prevent “leakage” of future words. During decoding, the attention scores for future words are masked, so the model can only focus on past and present words. Finally, the decoder’s output is processed through a linear layer and softmaxed to produce the final predicted probabilities.

Figure 9 represents the architecture of the vision transformer, where the vision transformer uses the encoding part of the transformer. The input image is broken into small patches. In the proposed work, input images are broken into 64 patches with 8 × 8 dimension, although in the figure, images are broken into 16 patches only just for simple and better understanding. The patches are then embedded with a flattened representation for further processing.

Unlike traditional neural networks, this model does not have a built-in understanding of the position of each sample in the sequence—each sample is a patch from the image. The image is processed along with a learnable positional embedding and passed into the encoder. The positional embeddings are trained during the process, rather than being predefined. Similar to BERT, a special token is included at the start. Each image patch is flattened into a vector and then multiplied by a learnable embedding matrix to create embedded patches. These embedded patches are combined with the positional embeddings and fed into the Transformer. The key distinction is that, instead of using a decoder, the encoder’s output is directly passed into a feed-forward neural network to generate the classification output.

Table 4 presents the number of parameters and shape of the ViT-B16 model. There are a total of 85,772,900 parameters counted, among which 85,771,108 parameters are considered as trainable. Figure 10 shows the layer-based flow diagram where the architecture consists of four dense layers.

### 3.5. Optimization and Hyperparameter Tuning

Optimization is important to reduce the model complexity as well as eliminate issues like overfitting and underfitting. To get maximum accuracy with minimum loss, hyperparameter tuning plays important roles where each parameter has to be considered carefully, assigning the most optimum values. However, it is difficult to assume which values should be appropriate for which parameters, and here the issue arises. The entire process follows a hit-and-trial approach where each value should be taken and tested the result accordingly. It would be a time-consuming process to check all parameters with all the possible values. Here the optimization algorithm comes that extends the ability to tune the model automatically rather than checking the value manually to get maximum output. Here in this work, the Grasshopper optimization algorithm has been utilized to find the best possible values for certain parameters. The parameters used in the work are batch size, epoch number, dropout rate, learning rate, and patch size. The lower bound and upper bound values for learning rate, dropout rate, batch size, and epoch are

Lower bound, LB = [0.00001, 0.1, 16, 5]

Upper bound, UB = [0.01, 0.6, 128, 30]

Best fitness value represents the optimum solution during the optimization process that finds the minimum or maximum values of the objective function depending on the minimization or maximization problem.

The proposed problem is a minimization one where the smallest fitness value is considered the best. Here the fitness value is calculated with the following formula:Fitness value = Validation loss − 0.1 × Validation accuracy

To develop vision transformer-based models, patch size needs to be mentioned. Figure 11 shows an input MRI image with respective patch images of (8, 8) where a total of 64 patches are there. Number of patches can be calculated through the following formula:Number of patch size = (Size of input image // patch size)

Table 5 presents the detailed parameters of each model used in the proposed framework development. From Table 5, it can be observed that the epoch has been taken to 10, 16, 20, and 25, and the learning rate has been taken to 0.001, 0.01, 0.001, and 0.00001 for ViT-B16, VGG19, Reset153V2, and EfficientNetV2B3.

### 3.6. Weighted Average Ensemble Technique

Weighted Average Ensemble (WAE) involves integrating trained models and assigning weights based on their performance to enhance the system’s overall accuracy. Four models—ViT-B16, VGG19, ResNet152V2, and EfficientNetV2B3—were individually trained, and their outputs were later combined using weighted values (WV1, WV2, WV3, and WV4) ranging from 0.1 to 0.4. The weights were determined based on the performance priority of each model, with the best-performing model receiving the highest weight. The predictions of each model were weighted through a tensor dot operation to produce the final combined output.

### 3.7. Systematic Designing Approach

The proposed framework has been developed through the combination of three CNN models and 1 ViT. Figure 12 shows the flow diagram for each step of the proposed framework. The very first step is to load data instances from the selected dataset. Once all the data are loaded, preprocessing has to be performed. Preprocessed data has to be split into two parts—train data and test data. Training and testing data are used for training and testing the models. Four models were trained separately with the trained data first. The Grasshopper optimization algorithm helps to find out the best parameters to fetch the best performance. Accordingly, the performance of each model, weight values—WV1, WV2, WV3, and WV4—have been assigned for four models M1 (vit-16), M2 (VGG19), M3 (ResNet152V2), and M4 (EfficientNetV2B3), respectively. After assigning weights, the ensemble technique is applied, where the combination of the performance of each model is utilized. Finally, validation of the ensemble model is performed with test data, and the classification result is obtained as the output.

### 3.8. Proposed Algorithm

IS: image size, DS: brain MRI Alzheimer dataset, E: number of epoch, BS: batch size, NCL: number of classes, TP: train folder path, PS: patch size, NPS: number of patch size, M1: ViT_b16, M2: VGG19, M3: ResNet152V2, M4: efficientNetV2B3, OPT: optimizer, lb: lower bound, ub: upper bound, LR: learning rate, AV: activation function (Algorithm 1).
**Algorithm 1**. DL-based Alzheimer’s disease detection approach using brain MRI images
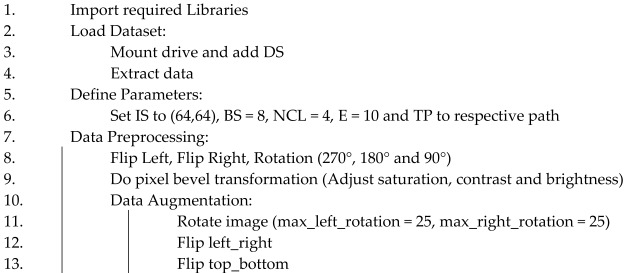

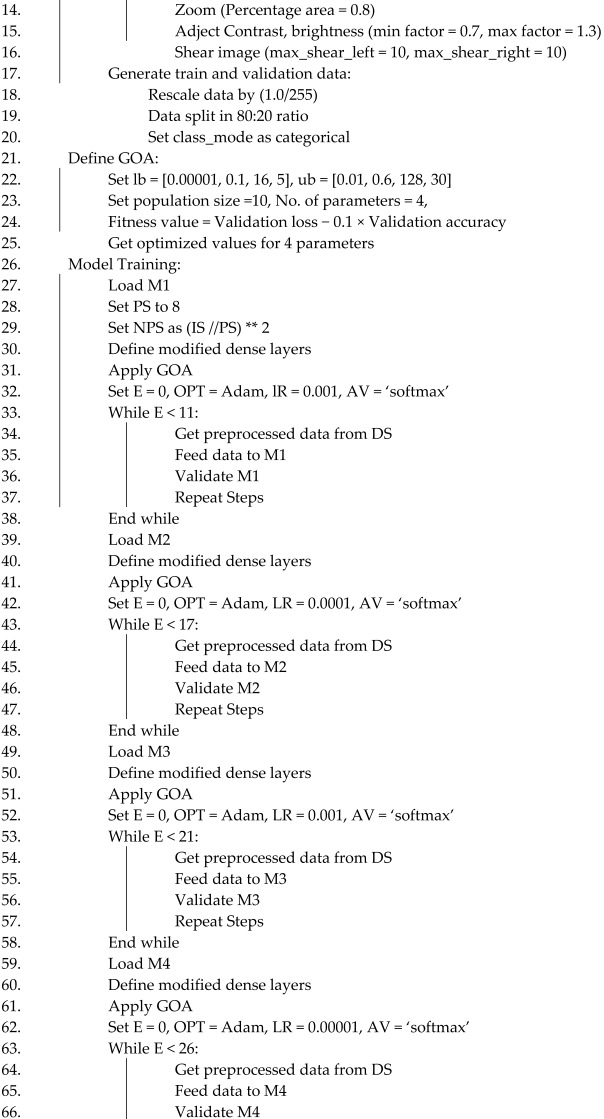

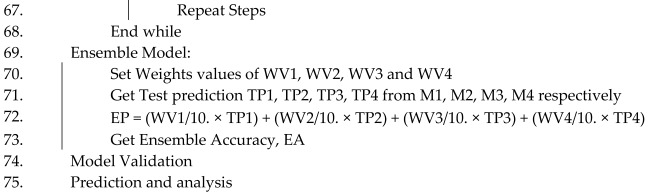


### 3.9. IoMT Infrastructure with Proposed Classification Framework

The Internet of Medical Things or IoMT, is a network of devices to access healthcare information through the internet. Data from IoMT devices can be shared remotely to anywhere, processed online, and accessed wirelessly. IoMT plays a significant role in telemedicine, where one can get healthcare services remotely.

Figure 13 shows a distributed network connected with a central cloud server, which is accessible to each user through the internet. The cloud server is integrated with an AI module where our proposed algorithm is defined. If any MRI images are received from any sources to the cloud server, it can classify the images instantly and send the images to the respective healthcare facilities accordingly. To make the scenario simple, three cases of data-sending sources have been shown and mentioned in the figure—MRI images from the hospital, MRI images from Personal IoMT devices, and MRI images from diagnostic labs. Thus, in any case, an MRI image is generated and sent to the main cloud server through the internet. The image is received and processed according to the mentioned steps shown in the AI module in the figure and classified into a level of dementia. The AI module is integrated with our proposed classification framework that is able to classify MRI images to find the dementia level. Firstly, the received data is resized as the model’s input size of our proposed classification framework. Then, the processed data is fed to the proposed classification model and generates the targeted classification output. Based on the classification report, images with the report will be sent to the respective healthcare persons, research facilities, or hospitals. Responses or feedback data received from the hospitals, research labs, or individual doctors are delivered to the respective end user accordingly. With the proposed infrastructure, one can get to know the level of dementia as well as get proper information or advice from physicians remotely.

## 4. Results and Discussion

In this section, the results of each experiment have been analyzed and properly presented to support the strength and novelty of the proposed approach. A brief discussion is presented based on the observation, and the comparative analysis has been performed with the existing state-of-the-art technique.

Table 6 presents the training accuracy, training loss and validation accuracy, and validation loss with each epoch. All models, including CNN and vit are optimized with GOA, where the best performances were obtained with the best parameter values. Table 7 shows the optimum parameters’ values to achieve maximum performance. Figure 14 presents confusion matrices individually of all the models used in the proposed framework, where Figure 15 shows the confusion matrix of the ensemble model.

Table 8 shows the performance metrics individually for each class, and Figure 16 shows the overall performances through a bar graph.

Table 9 shows the resulting values of training and testing accuracies for each model, where it is easily identifiable that the ensemble model with optimization comes with better performance with 97.31% accuracy. The processing time during the training is also less than in other models, which ensures the time efficacy of the proposed approach. Table 10 shows weight distribution, where four weight values (WV) have been assigned to all the four models (ViT-B16, VGG19, ResNet152V2, and EfficientNetV2B3) to combine performances. At the very first combination, it is seen that WV1 For ViT-B16 got a low value, and WV4 for EfficientNetV2B3 received high values where ViT-B16 performed much better than EfficientnetV2B3. The resultant ensemble accuracy was not increased much for the same reason. At the last attempt, WV1 got a high weight value, and WV4 got a low value, presenting the perfect combination of weights where maximum ensemble accuracy was obtained.

Table 11 shows the 5-fold cross-validation method for analyzing the consistency of the performance. The data have been divided into five bins, and for each attempt, each bin is applied to check the result. After getting five different accuracies for each model, mean, variance, and standard deviation are calculated to check the data variability. Table 12 shows the final performance metrics obtained with test data, where accuracy, precision, recall, and F1 score are given. Table 13 shows the accuracy and loss status of the proposed ensemble model during training and validation with and without the optimization technique. It is clearly observed that at the 10th epoch, maximum accuracy and minimum loss are achieved with the optimization technique.

Table 14 shows the comparative analysis between the proposed work and the state-of-the art work.

Figure 17 shows the accuracy and loss graph, where the first plot shows the training and validation accuracy, and the second plot shows the train and validation loss with the epoch increment. Generally, loss gets decreased and accuracy gets increased with epoch increment [83]. Initially, loss was very high and accuracy was very low. After increasing the epoch number, accuracy increased and loss decreased. At the 10th epoch, the maximum accuracy with minimum loss was recorded. ViT-B16 is a heavyweight model that requires a lot of computational time with complex operations. To utilize training time, an early stopping function was applied that stops the epoch iteration at a value of 10, which assumes that at the 10th epoch, models were well trained to perform with good accuracy and less loss.

Figure 18 shows the accuracy and loss performance obtained by the VGG19 model, where at the 16th epoch, the highest accuracy and lowest loss were received. In the first plot, training accuracy performs better than validation accuracy, which indicates issues like model overfitting. In Figure 19, it goes to the 20th epoch and records 89% train accuracy with 0.7% train loss.

In Figure 20, the maximum train accuracy of 84% was received at the 25th epoch, where the minimum loss was received at the same epoch value. Although the training and validation curves in this figure show consistent performance, both the train and validation accuracies achieved at the 25th epoch are lower than those of other models.

Observing the performance from Figure 18, Figure 19 and Figure 20, it is found that both VGG19 and ResNet152V2 offer quite similar performance, where EfficientNetV2B3 comes with lower accuracy for both training and validation.

Figure 21 represents the accuracy loss graph of the proposed ensemble model during training and validation. The comparison has been drawn between optimized and nonoptimized performance, where the optimized ensemble model comes with promising performance, ensuring minimum loss and maximum accuracy.

## 5. Conclusions

In conclusion, despite being a severe public health threat, AD has a complex pathology and remains challenging for memory, cognition, and behavior. Early detection and intervention, most importantly in the MCI stage, is the best chance to slow the disease’s progression. Applications of ML and DL via the analysis of multimodal data, such as neuroimaging data like MRI and CT scans, have shown promising results for improving diagnostic accuracy. Besides helping to detect such diseases early, these technologies also lead to more personalized and effective interventions. To some extent, these are accomplished, but there remains much to do, especially in developing interpretable and simple-to-deploy AI models in the clinical setting. However, by overcoming these hurdles and further fine-tuning predictive models, we could introduce AI in AD care to transform patient care. Through ongoing research and collaboration, these AI-driven tools can help create better outcomes for those with AD and reduce the burden that this disease potentially can cause. In this work, an optimized classification framework was developed, applying an ensemble technique, utilizing the power of vision transformers and three CNN models. The work offers promising performance, including high accuracy of 97.31%, precision of 97.32, recall of 97.35, and F1 score of 0.97. However, issues like image overlapping may cause feature redundancy where CNN models cannot identify the unique spatial features from images. The study relies solely on secondary data collection, whereas incorporating primary data collection could help address the limitations of the classification framework and enhance its robustness. This study can be found valuable for AI-based healthcare research, wherein, in the future, it can put important significance on the further development of a neuroimaging-based classification approach with novel and innovative features. Embedded devices can be implemented and integrated with MRI machines, through which classification can be performed at the same time as image generation.

## Figures and Tables

**Figure 1 diagnostics-15-00789-f001:**
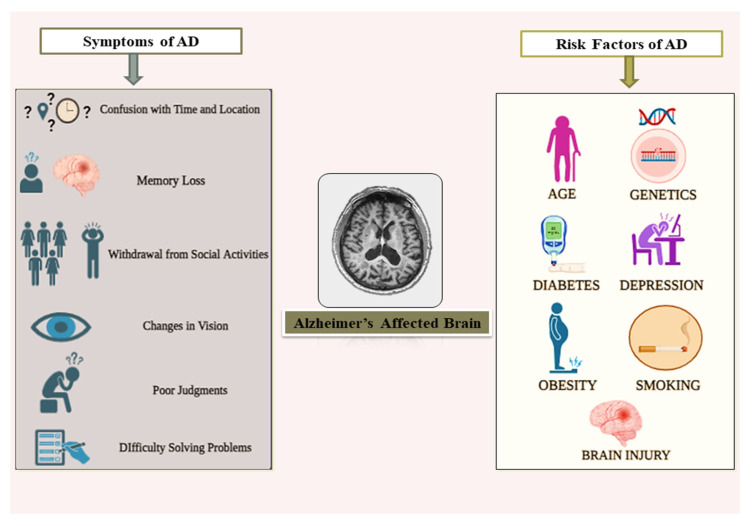
An overview of symptoms and risk factors in AD.

**Figure 2 diagnostics-15-00789-f002:**
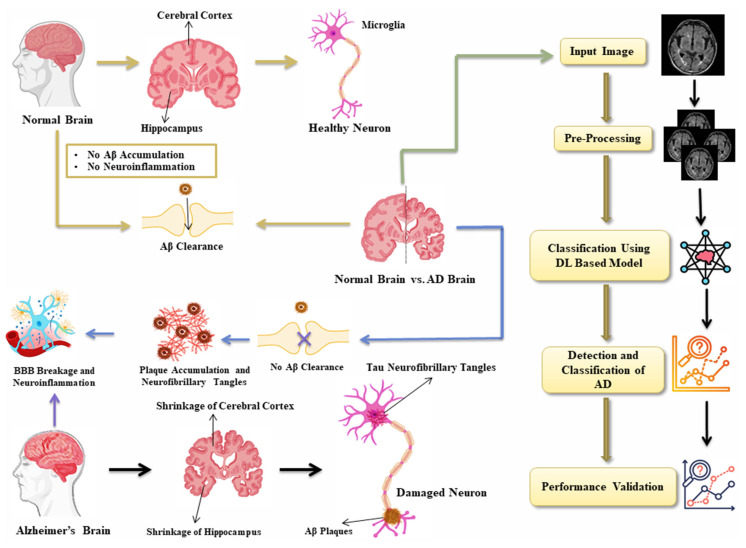
Pathophysiology and detection strategies in AD.

**Figure 3 diagnostics-15-00789-f003:**
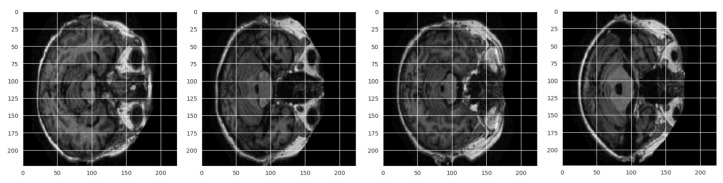
Sample instances from all available classes of OASIS Alzheimer’s Detection dataset.

**Figure 4 diagnostics-15-00789-f004:**
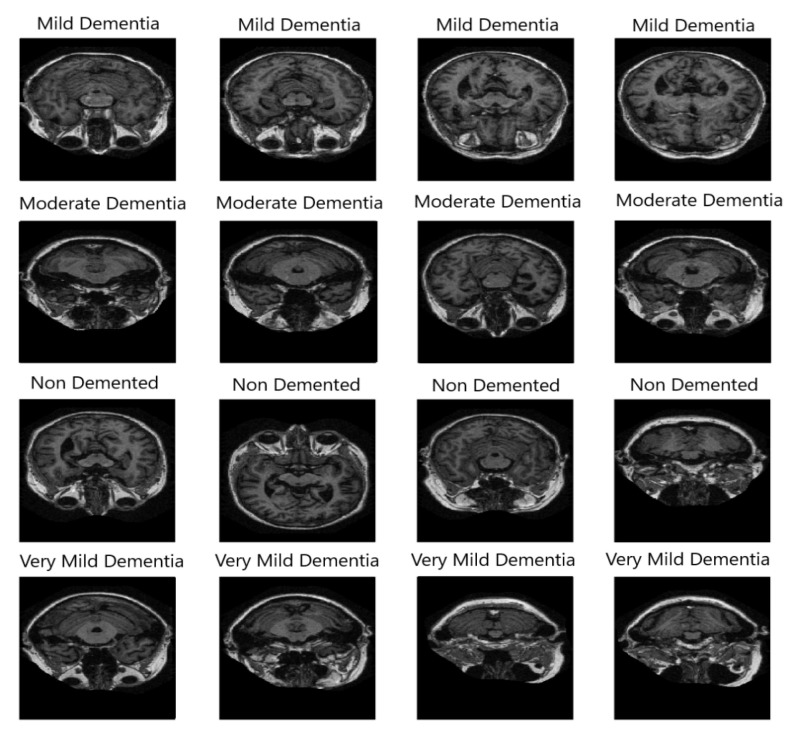
Preprocessed image samples indicating respective class name.

**Figure 5 diagnostics-15-00789-f005:**
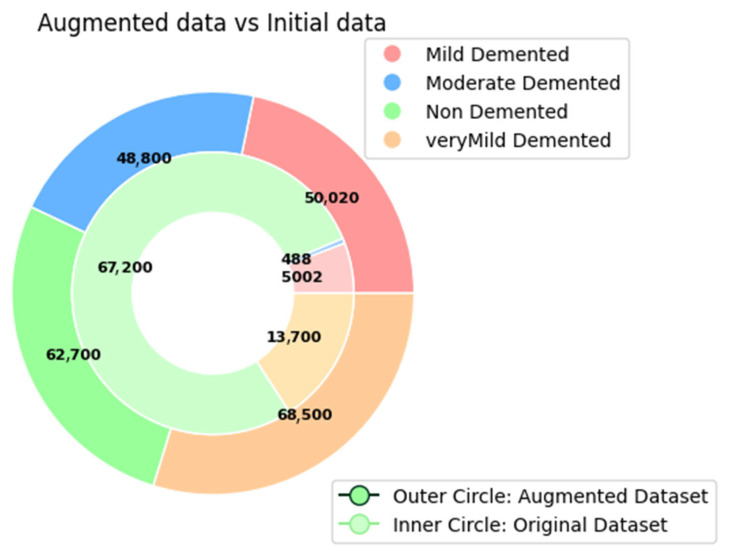
Dataset description, before and after data augmentation.

**Figure 6 diagnostics-15-00789-f006:**
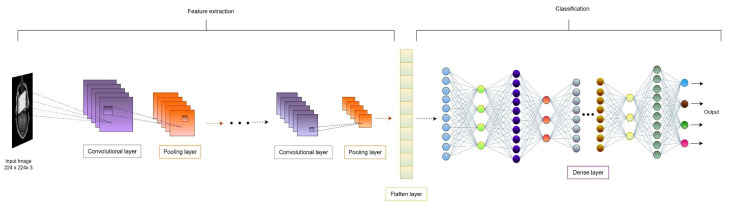
Basic architecture of convolutional neural network.

**Figure 7 diagnostics-15-00789-f007:**
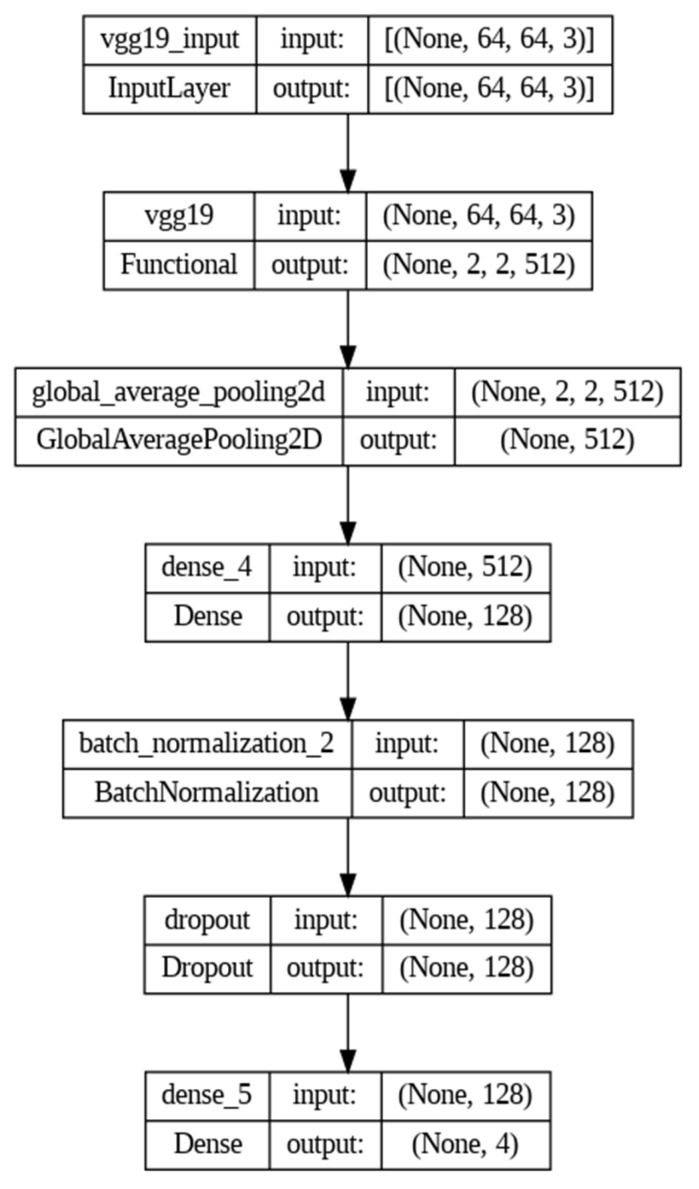
Layer-based architecture of the VGG19 model used in the proposed framework.

**Figure 8 diagnostics-15-00789-f008:**
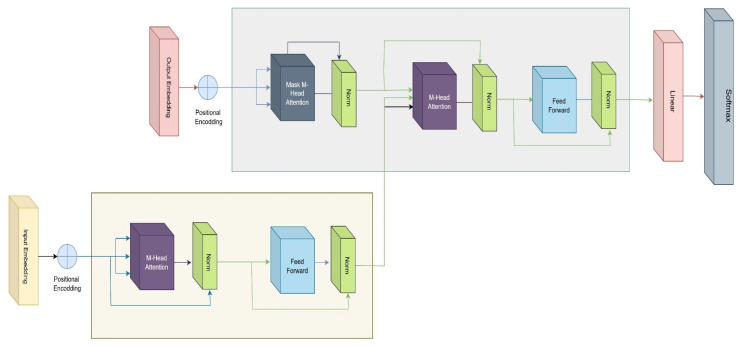
Architecture of a basic transformer.

**Figure 9 diagnostics-15-00789-f009:**
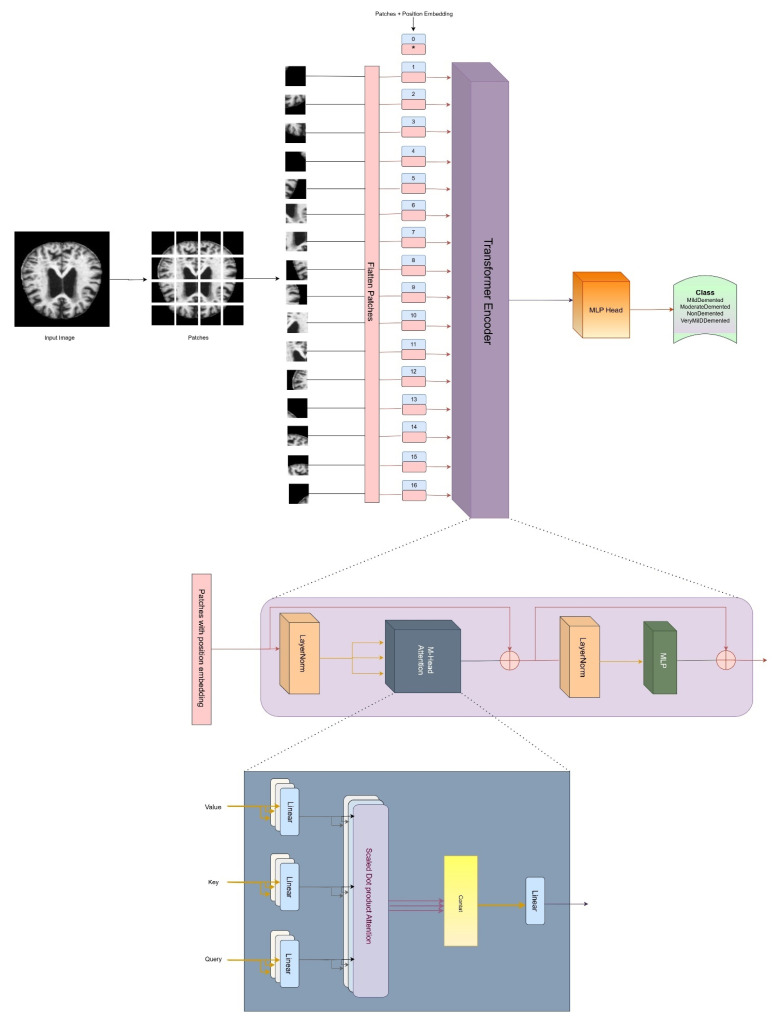
Functional architecture of vision transformer (ViT).

**Figure 10 diagnostics-15-00789-f010:**
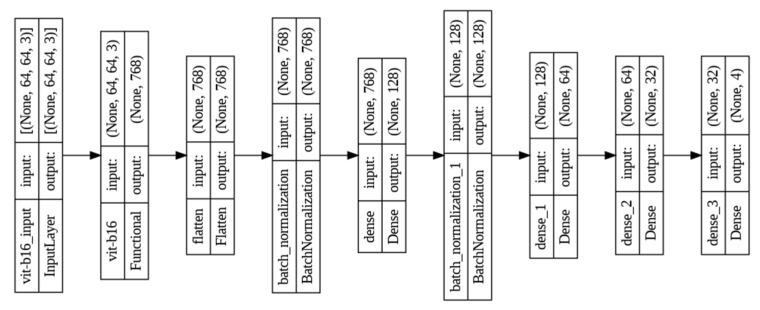
Layer-wise architecture of ViT-B16 model.

**Figure 11 diagnostics-15-00789-f011:**
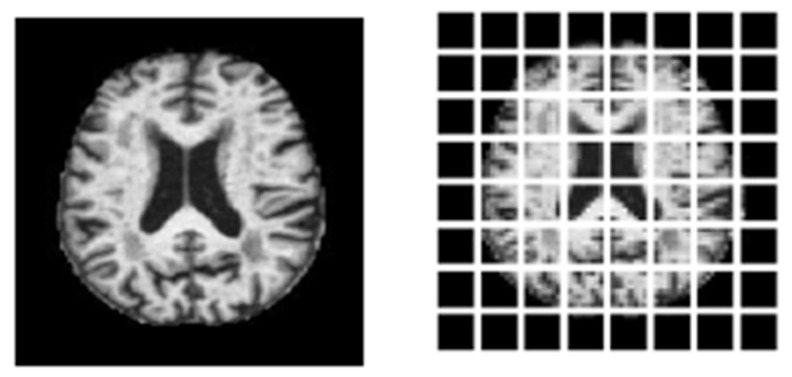
Brain MRI image with respective 64 patches.

**Figure 12 diagnostics-15-00789-f012:**
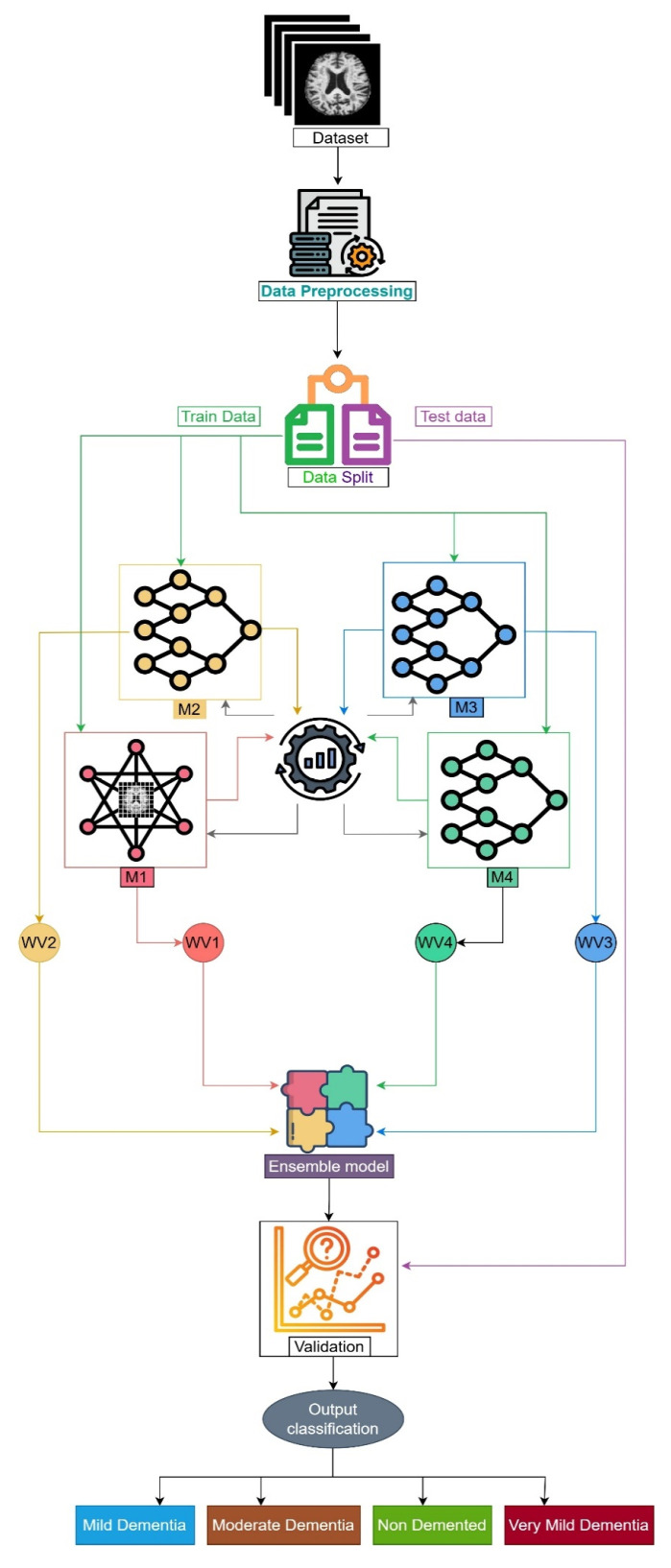
Workflow diagram of the proposed framework.

**Figure 13 diagnostics-15-00789-f013:**
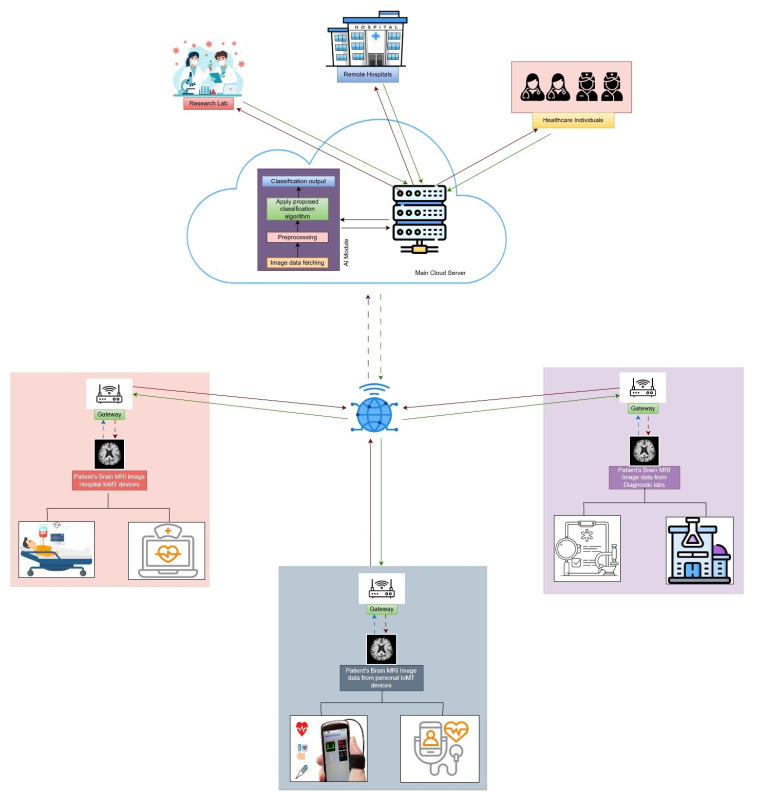
Cloud-AI-based IoMT infrastructure for remote diagnosis of Alzheimer disease.

**Figure 14 diagnostics-15-00789-f014:**
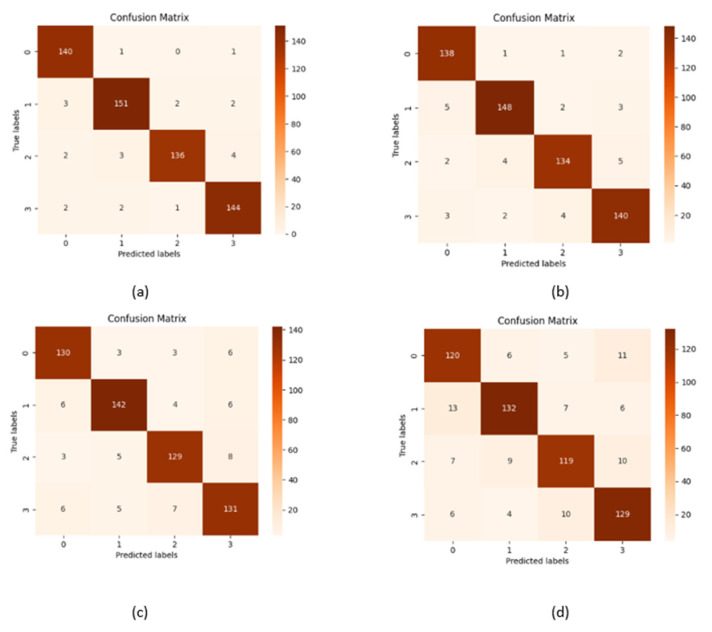
Confusion matrix of (**a**) ViT-B16, (**b**) VGG19, (**c**) ResNet152V2, and (**d**) EfficientNetV2B3.

**Figure 15 diagnostics-15-00789-f015:**
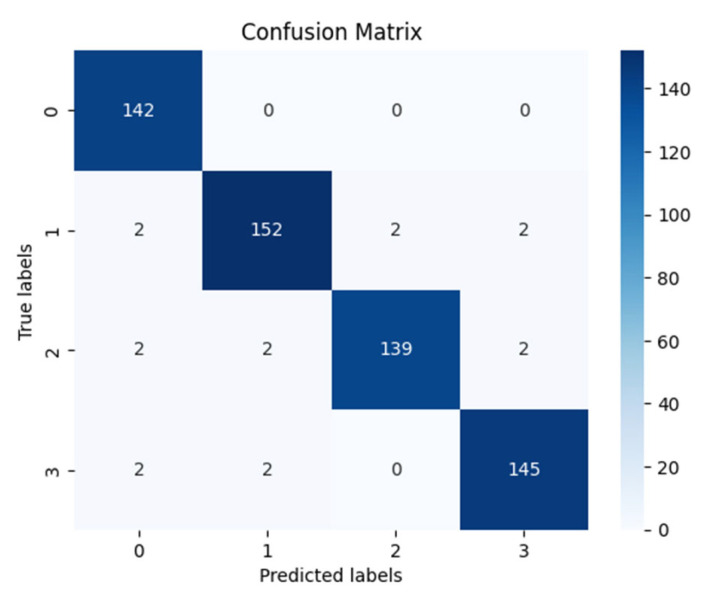
Confusion matrix obtained by the weighted average ensemble technique.

**Figure 16 diagnostics-15-00789-f016:**
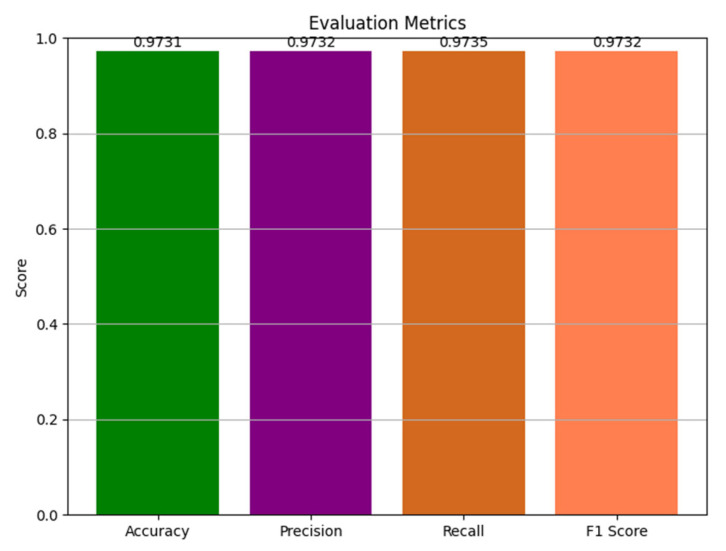
Performance metrics with a weighted average ensemble approach.

**Figure 17 diagnostics-15-00789-f017:**
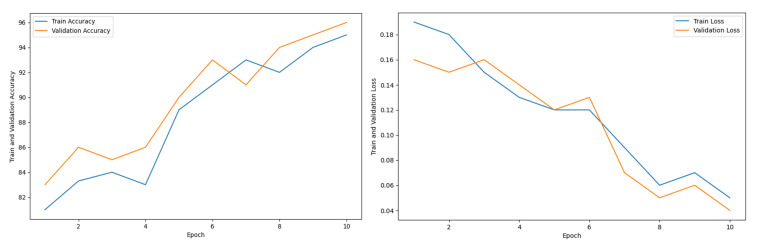
Accuracy and loss graph for ViT-B16.

**Figure 18 diagnostics-15-00789-f018:**
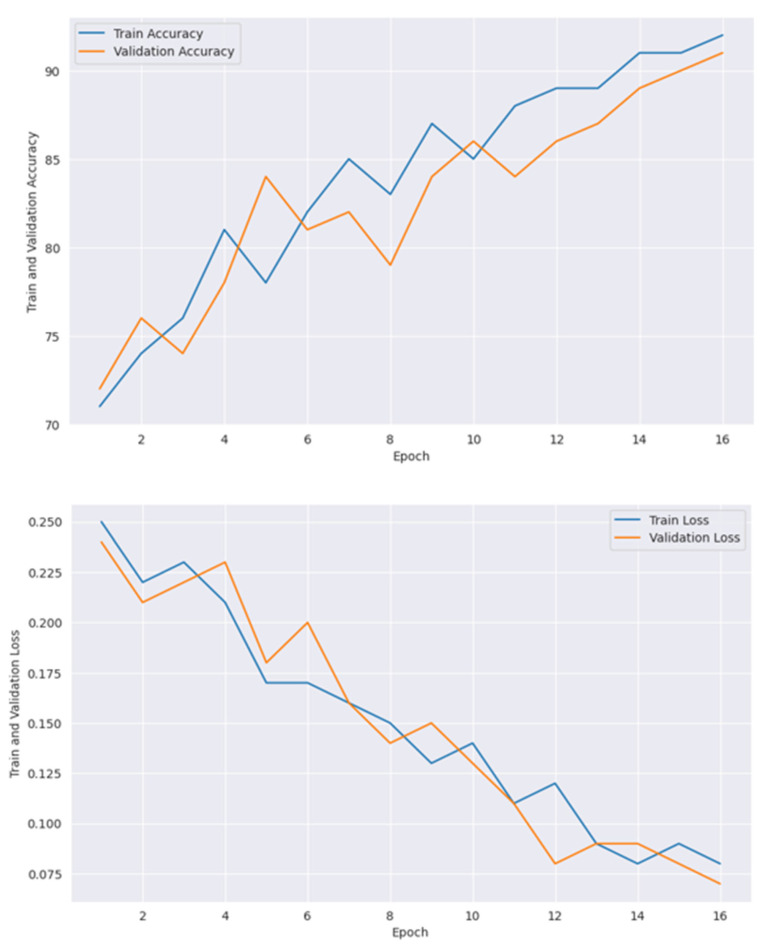
Accuracy and loss graph of VGG19.

**Figure 19 diagnostics-15-00789-f019:**
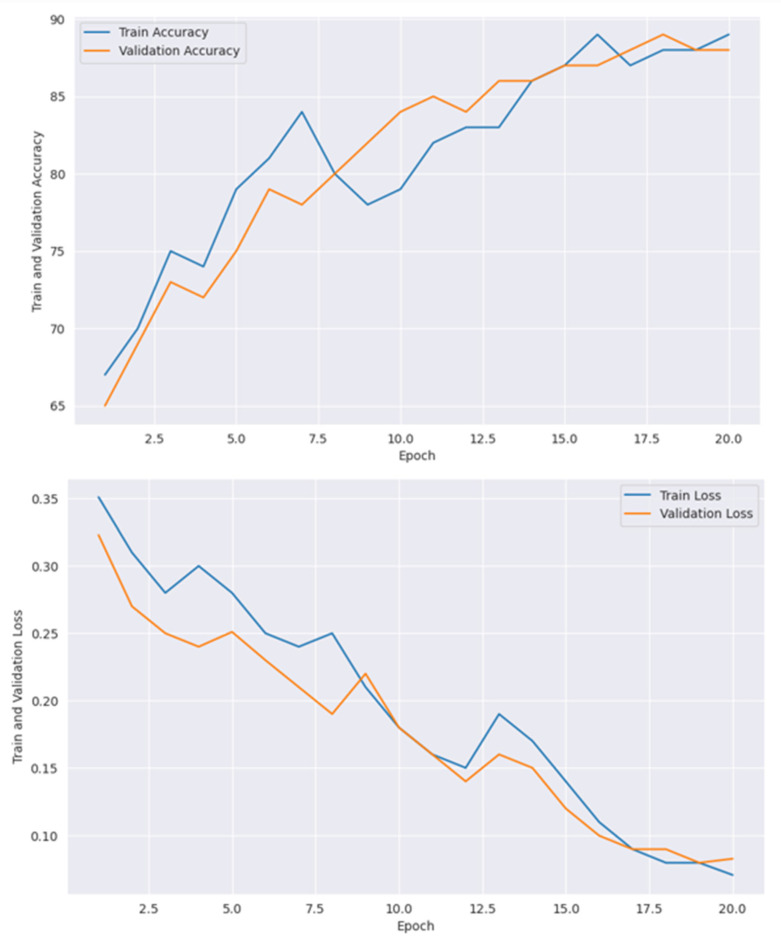
Accuracy and loss graph for ResNet152V2.

**Figure 20 diagnostics-15-00789-f020:**
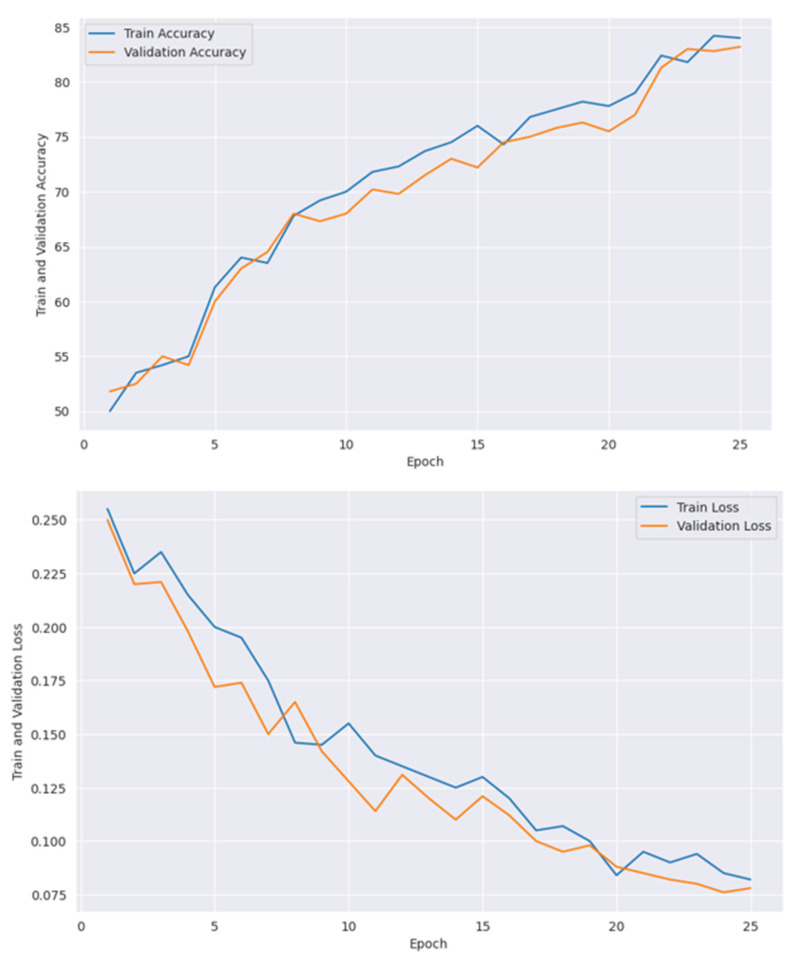
Accuracy and loss graph of EfficientNetV2B3.

**Figure 21 diagnostics-15-00789-f021:**
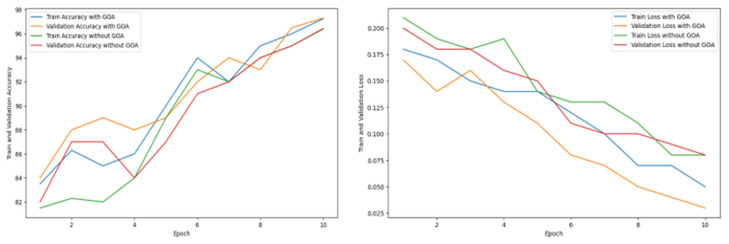
Accuracy and loss status of the proposed model during training and validation.

**Table 1 diagnostics-15-00789-t001:** Summary of Related Works.

Author (s)	Model Used	Dataset	Total Images	Methodology	Accuracy	Focus Area	Limitations
Qu et al. (2023) [21]	UNB-GCN	ADNI Dataset	928	Used univariate neurodegeneration biomarkers with GCN.	93.90% (CU vs. AD) 82.05% (MCI vs. AD)	Early AD detection	Lacks multimodal validation.
Liu (2023) [39]	DenseNet -AM	ADNI Dataset	413	Integration of hippocampal and whole-brain MRI	92.8%	Structural abnormality analysis	Limited feature diversity
Han et al. (2021) [40]	LightGBM	Real Datasets (EADC and ADNI Dataset)	78 (41 NC and 37 AD) * 19 Features = 1482 Images	Feature optimization for improved classification	89.7% (AD Classification Feature) 82.1% (Hippocampal Volume)	AD diagnosis	Inconsistent performance on small datasets
Sanjeev Kumar K et al. (2023) [42]	InceptionResNetV2 and ResNet50	ADNI dataset	6400 MRI Images	Deep learning architecture for AD stage classification	86.84% 90.27%	Staged AD classification	Requires high-quality annotated data
Mishra et al. (2024) [43]	ML models (LR, RF, SVM, DT, AdaBoost)	OASIS Project Dataset	150 Subjects	Analysis of early prediction problems in AD	86.84% (RF) (Highest)	Early prediction	Data inconsistency across datasets
Lu et al. (2023) [44]	ConvNeXt-DMLP	ADNI Dataset	33,984	Multiclassification framework to reduce imaging overlap of AD and MCI	78.95%	AD and MCI classification	Limited generalization to external datasets
Prajapati et al. (2021) [45]	Deep Neural Networks (DNN)	ADNI Dataset	58 AD, 48 MCI and 73 CN Subjects	Binary classification using deep neural networks	85.19% (AD vs CN) 76.93% (MCI vs. CN) 72.73% (AD vs MCI)	AD stage classification	Limited scope for multiclass classification
Das et al. (2022) [49]	Hippocampal Segmentation with CNN	ADNI and OASIS Dataset (MRI)	100 people (Train) and 35 people (Test)	Focused on hippocampal segmentation for structural MRI	91.43%	Structural abnormality detection	Limited generalizability to other brain regions
Pallawi et al. (2023) [56]	EfficientNetB0	Dataset from Kaggle	6400 images (MD-896, MOD-64, ND-3200, and VMD-2240)	Transfer learning for early-stage AD classification	95.78%	Early-stage classification	Dataset imbalance
Islam et al. (2023) [57]	YOLO (YOLOv3, YOLOv4, YOLOv5, YOLOv6 and YOLOv7	ADNI Dataset (MRI data)	300 Subjects (2000+ Images)	Automated hippocampal detection for early recognition	YOLOv7 (81%-AD vs. MCI, 95%-AD vs. CN, 83%-CN vs. MCI)	Early-stage identification	Limited automation coverage
Yin et al. (2022) [34]	SMIL-DeiT	ADNI dataset	2032 Subjects	Staged AD classification with enhanced techniques	93.2%	Staged classification	Model interpretability
Jin et al. (2021) [76]	VAE+GAN	ADNI Dataset (sMRI Images)	1042 Subjects	Identification of brain atrophy	94%	Structural analysis	Overfitting potential

**Table 2 diagnostics-15-00789-t002:** Comparison of VGG19, ResNet152V2, and EfficientNetV2B3 Architectures for Image Classification.

Feature	VGG19	ResNet152V2	EfficientNetV2B3
Architecture Type	A deep CNN with a straightforward sequential design	A deep residual network incorporating skip connections	An optimized convolutional network with depth-wise convolutions
Depth (Layers)	19 layers	152 layers	Adaptive depth based scaling
Parameter Count	Approximately 143 million	Approximately 60 million	Nearly 24 million
Advantages	Simple yet effective for smaller datasets	Overcomes vanishing gradient issues and excels in deep feature extraction	Efficiency in computation while maintaining high accuracy
Limitations	High parameter counts leading to increased computational cost	More complex training process with longer inference time	Requires sophisticated training techniques for optimal results
Feature Extraction Capability	Captures fine-grained spatial details effectively	Excels at extracting deep semantic representations	Strikes a balance between feature learning and efficiency
Training Time	Relatively long	Moderate	Faster compared to the others
Accuracy Performance	Moderate to high	High	Very high
Reason for Selection	Included due to its ability to retain detailed spatial information	Chosen for its deep feature extraction and improved gradient handling	Selected for its efficiency, optimizing both accuracy and computational demands

**Table 3 diagnostics-15-00789-t003:** Shape and Parameter Size of “VGG19” Model.

Layer (Type)	Output Shape	Param #
vgg19 (Functional)	(None, 2, 2, 512)	20,024,384
global_average_pooling2d (GlobalAveragePooling2D)	(None, 512)	0
dense_4 (Dense)	(None, 128)	65,664
batch_normalization_2 (BatchNormalization)	(None, 128)	512
dropout (Dropout)	(None, 128)	0
dense_5 (Dense)	(None, 4)	516
Total params: 20,091,076		
Trainable params: 66,436		
Nontrainable params: 20,024,640		

**Table 4 diagnostics-15-00789-t004:** Shape and Parameter Size of “Vision Transformer (ViT-B16)” Model.

Layer (Type)	Output Shape	Param #
ViT-B16 (Functional)	(None, 768)	85,660,416
flatten_6 (Flatten)	(None, 768)	0
batch_normalization_12 (Batch Normalization)	(None, 768)	3072
dense_24 (Dense)	(None, 128)	98,432
batch_normalization_13 (Batch Normalization)	(None, 128)	512
dense_25 (Dense)	(None, 64)	8256
dense_26 (Dense)	(None, 32)	2080
dense_27 (Dense)	(None, 4)	132
Total params: 85,772,900		
Trainable params: 85,771,108		
Nontrainable params: 1792		

**Table 5 diagnostics-15-00789-t005:** Parameter details of developed models utilized in the proposed classification frameworks.

Parameters	Classification with ViT-B16	Classification with VGG19	Classification with ResNet152V2	Classification with EfficientNetV2B3
Train_loss	0.05	0.08	0.07	0.08
Train_accuracy	95.21	92.14	89.13	84.15
Training time	4 h 17 min	2 h 13 min	2 h 39 min	1 h 46 min
Epoch	10	16	20	25
Optimizer	Adam	Adam	Adam	Adam
Initial learning rate	0.001	0.0001	0.001	0.00001
Loss function	categorical cross entropy	categorical cross entropy	categorical cross entropy	categorical cross entropy
Output activation function	Softmax	Softmax	Softmax	Softmax

**Table 6 diagnostics-15-00789-t006:** Accuracy and Loss Values Obtained by the ViT-B16 Model.

Number of Epoch	Train Accuracy	Train Loss	Validation Accuracy	Validation Loss
1	81	0.19	83	0.16
2	83	0.18	86	0.15
3	84	0.15	85	0.16
4	83	0.13	86	0.14
5	89	0.12	90	0.12
6	91	0.12	93	0.13
7	93	0.09	91	0.07
8	92	0.06	94	0.05
9	94	0.07	95	0.06
10	95	0.05	96	0.04

**Table 7 diagnostics-15-00789-t007:** Tuned Parameters with GOA.

Models	Best Learning Rate	Best Dropout Rate	Best Batch Size	Best Number of Epochs	Highest Accuracy	Minimum Loss	Best Fitness Score
ViT-B16	0.001	-	8	10	95.21	0.05	−0.03
VGG19	0.0001	0.2	32	16	92.14	0.08	−0.01
ResNet152V2	0.001	0.2	16	20	89.13	0.07	−0.01
EfficientNetV2B3	0.00001	0.1	32	25	84.15	0.08	−0.004

**Table 8 diagnostics-15-00789-t008:** Classification Report of Ensemble Model.

	Precision	Recall	f1-Score	Support
Mild Demented	0.9595	1.0000	0.9793	142
Moderate Demented	0.9744	0.9620	0.9682	158
Nondemented	0.9858	0.9586	0.9720	145
Very Mild Demented	0.9732	0.9732	0.9732	149
accuracy			0.9731	594
macro avg	0.9732	0.9735	0.9732	594
weighted avg	0.9733	0.9731	0.9730	594

**Table 9 diagnostics-15-00789-t009:** Training Testing Accuracies and Processing Time Obtained by DL models.

Models	Training Accuracy (%)	Testing Accuracy (%)	Processing Time During Training
ViT-B16	95.21	96.12	4 h 17 m
VGG19	92.14	94.27	2 h 13 m
ResNet152V2	89.13	89.56	2 h 39 m
EfficientNetV2B3	84.15	84.17	1 h 46 m
Ensemble model (Not optimized)	96.39	96.43	1 h 55 m
Ensemble model (Optimized with GOA)	97.25	97.31	1 h 41 m

**Table 10 diagnostics-15-00789-t010:** Weight Value Distribution with Respective Obtained Accuracy.

WV1	WV2	WV3	WV4	Ensemble Accuracy
0.1	0.2	0.3	0.4	96.42
0.2	0.1	0.4	0.3	96.68
0.3	0.4	0.1	0.2	96.83
0.4	0.3	0.2	0.1	97.19
0.4	0.3	0.3	0.4	97.04
0.4	0.3	0.4	0.3	97.25
0.4	0.3	0.3	0.1	97.31

**Table 11 diagnostics-15-00789-t011:** Five-fold cross-validation method for performance analysis of different classification frameworks.

Split Count	ViT-B16	VGG19	ResNet152V2	EfficientNetV2B3	Ensemble Model
1	96.04	94.22	89.51	84.11	97.26
2	95.86	94.14	89.36	84.04	97.14
3	95.96	94.08	89.22	83.99	96.99
4	96.08	94.18	89.41	83.94	97.12
5	95.88	94.05	89.31	84.09	97.06
Arithmetic Mean	95.96	94.13	89.36	84.03	97.14
Variance	0.007	0.003	0.009	0.003	0.008
Standard deviation (std.)	0.08	0.06	0.09	0.06	0.08

**Table 12 diagnostics-15-00789-t012:** Performance Metrics Obtained from the MRI Image-Based Classification Framework.

Performance metrics	ViT-B16	VGG19	ResNet152V2	EfficientNetV2B33
Accuracy	96.12	94.27	89.56	84.17
Precision	96.15	94.27	89.55	84.18
Recall	96.14	94.30	89.57	84.17
F1 Score	0.9613	0.9427	0.8955	0.8414

**Table 13 diagnostics-15-00789-t013:** Performance Status of Proposed Ensemble Model During Training and Validation.

Epoch	Train Accuracy with GOA	Validation Accuracy with GOA	Train Accuracy Without GOA	Validation Accuracy Without GOA	Train Loss with GOA	Validation Loss with GOA	Train Loss Without GOA	Validation Loss Without GOA
1	83.51	84.37	81.54	82.73	0.18	0.17	0.21	0.20
2	86.34	88.54	82.38	87.58	0.17	0.14	0.19	0.18
3	85.54	89.86	82.28	87.52	0.15	0.16	0.18	0.18
4	86.68	88.36	84.56	84.87	0.14	0.13	0.19	0.16
5	90.67	89.34	89.25	87.23	0.14	0.11	0.14	0.15
6	94.68	92.75	93.86	91.78	0.12	0.08	0.13	0.11
7	92.36	94.97	92.64	92.64	0.10	0.07	0.13	0.10
8	95.43	93.65	94.16	94.34	0.07	0.05	0.11	0.10
9	96.28	96.52	95.31	95.78	0.07	0.04	0.08	0.09
10	97.25	97.31	96.39	96.43	0.05	0.03	0.08	0.08

**Table 14 diagnostics-15-00789-t014:** Comprehensive Comparison with Other Existing Studies.

Authors	Model Used	Dataset	Total Images	Methodology	Accuracy	Focus Area	Limitations
Butta et al. (2023) [80]	Ensemble DL Model (O-ANN, Capsule CNN, Autoencoder)	Kaggle Dataset	6400 MRI Images	Preprocessing (histogram equalization, Gaussian filtering, skull stripping), ROI extraction (attention-based U-NET with EEO optimization)	94.52%	Early and precise AD diagnosis using MRI	Computational complexity due to hybrid optimization and ensemble techniques; limited testing on diverse datasets
Agarwal et al. (2023) [81]	EfficientNet-b0 CNN with fusion of end-to-end and transfer learning (TL)	IXI and ADNI Dataset	600 MRI scans 916 MRI Scans (ADNI)	Preprocessing pipeline applied to T1W MRI scans; CNN model trained and evaluated for sMCI vs. AD (binary) and AD vs. CN vs. sMCI (multiclass) classification	Binary: 93.10% (testing), Multiclass: 87.38% (testing)	Early detection and staging of AD using MRI biomarkers	Requires further validation on larger and more diverse datasets; computational demands of CNN training
Saim et al. (2024) [82]	Nine CNN architectures (AlexNet, Inception, EfficientNet-b0, etc.)	Kaggle Alzheimer’s dataset ADNI dataset	5091 Images (Train) 1273 Images (Test)	Feature extraction using pretrained CNNs (fully connected, inception-based, residual-based, etc.); classification using MSVM, KNN, DT;	Kaggle: ADNI: 78.54% (MSVM)	Early detection and staging of AD using neuroimaging biomarkers	Performance on ADNI dataset lower; needs further validation on larger and more diverse datasets.
Sanjeev Kumar et al. (2023) [42]	InceptionResNetV2 and ResNet50	ADNI dataset	6400 MRI Images	Deep learning architecture for AD stage classification	86.84%	Staged AD classification	Requires high-quality annotated data
Lu et al. (2023) [44]	ConvNeXt-DMLP	ADNI Dataset	33,984	Multiclassification framework to reduce imaging overlap of AD and MCI	78.95%	AD and MCI classification	Limited generalization to external datasets
Proposed Work	Weighted ensemble model combined of ViT-B16, VGG19, ResNet152V2, EfficientNetV2B3	OASIS Dataset, collected from kaggle	234,220	Multiclass classification framework, optimized with Grasshopper Optimization Algorithm	97.31%	Classify dementia status that can be integrated to IoMT infrastructure	

## Data Availability

The data presented in this study are openly available in Kaggle “OASIS Alzheimer’s Detection” at https://doi.org/10.1162/jocn.2007.19.9.1498. “Data were provided 1-12 by OASIS-1: Cross-Sectional: Principal Investigators: D. Marcus, R, Buckner, J, Csernansky J. Morris; P50 AG05681, P01 AG03991, P01 AG026276, R01 AG021910, P20 MH071616, U24 RR021382”.

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
