# Peer review of "A Novel Diagnostic Framework with an Optimized Ensemble of Vision Transformers and Convolutional Neural Networks for Enhanced Alzheimer’s Disease Detection in Medical Imaging"

_diagnostics, 2025, doi:10.3390/diagnostics15060789_

Round 1

Reviewer 1 Report

Comments and Suggestions for Authors

Comment 1: The Abstract presented a well-structured and informative summary of the research work, with clear objectives, methods, and results. However, the abstract needs some formatting regarding the use of acronyms. For instance: AI, CNN, VGG, MRI. I suggest the authors indicate the database where the dataset was acquired not to mention that the proposed model was trained trained with 0.234 million brain MRI images.

Comment 2: The authors discussed the general importance of AI, ML, and DL in AD diagnosis and mention challenges like interpretability. It is very important for the authors to explain why CNN alone is seen as insufficient, explain why ViTs are beneficial, and justify the ensemble approach towards the conclusive part of the introduction i.e before stating the contributions of the proposed model

Comment 3: The authors currently present an image from only one class in the dataset. To enhance clarity and provide a more comprehensive understanding of the dataset, I recommend including representative images from all classes. This will allow readers to better appreciate the variations between different cognitive states. The resolution of the image needs to be improved.

Comment 4: Improve the resolution of Figure 4 - Figure 16. Confusion matrix in Fig 14 needs to be improved.

Comment 5: Table 12 looks confusing. The authors should represent the rows and columns properly.

Comment 6: What is the processing time of each of the models compared to the proposed ensemble model. This comparison will help determine whether the ensemble model provides a significant improvement in accuracy at a reasonable computational cost, or if it introduces trade-offs in terms of speed and resource consumption.

Comment 7: The subsection "3.5. Optimization and Hyperparameter Tuning" could be presented more clearly and structured better to enhance technical depth. The explanation lacks mathematical rigor and does not formally define the optimization process.

Comment 8: The authors have not provided the training loss and accuracy metrics for the proposed ensemble model. Including these metrics is essential for assessing the model's performance, convergence, and generalization ability. The authors used the Grasshopper Optimization Algorithm in the proposed model. Showing the graph of loss and accuracy both without optimization and with the optimization technique will help justify its effectiveness. This comparison will demonstrate the impact of the optimization algorithm on model performance, helping readers understand how it improves convergence, reduces overfitting, or enhances classification accuracy. I recommend including these visualizations and a discussion on how the optimization method specifically benefits the model in the context of Alzheimer’s Disease diagnosis. The authors should present a clear comparison of training and validation accuracy as well as training and validation loss between the optimized and unoptimized models. This comparison should highlight the effectiveness of the Grasshopper Optimization Algorithm in finding the optimal values for hyperparameters (learning rate, batch size, epoch number, dropout rate, etc.) by showing improved convergence or better generalization

Comment 9: The authors should also highlight the time efficiency of the Grasshopper Optimization Algorithm in terms of training time and inference time when applied to their ensemble model.

Comment 10: The authors should explain how the optimization algorithm helps to automatically fine-tune hyperparameters such as learning rate, batch size, dropout rate, and epoch number.

Comment 11: The authors may briefly discuss the potential limitations of the proposed method and what are the future research directions of this study.

Author Response

We sincerely appreciate for the time and effort to provide insightful and constructive feedback on our manuscript. Your valuable suggestions have significantly helped us refine and enhance the quality of our work. We have carefully addressed each comment and made the necessary revisions to improve the clarity, structure, and technical depth of our manuscript. Your guidance has been instrumental in strengthening our study, and we are grateful for your support in making this research more impactful. Thank you for your thoughtful evaluation and for helping us improve our work. All the corrections/modifications are highlighted in yellow marks on the revised manuscripts for your better understanding.

  1. We appreciate your positive remarks on the structure and clarity of our abstract. We have revised the abstract to ensure proper formatting of acronyms such as AI, CNN, VGG, and MRI, introducing them appropriately upon first mention. Additionally, we have explicitly stated the source of the dataset to provide clarity regarding its acquisition. Moreover, we have reviewed and corrected the phrasing related to the dataset size to eliminate redundancy and enhance readability.
  2. In the revised manuscript, we have expanded the introduction to clearly explain the challenges associated with CNNs, particularly addressed the limitation of CNN architecture, which may limit their ability to capture long-range dependencies in medical imaging. Additionally, we have highlighted the benefits of ViTs, including their self-attention mechanism, which enables a more holistic analysis of spatial relationships within images. Furthermore, we have justified the adoption of an ensemble approach, emphasizing how it combines the strengths.
  3. We acknowledge the importance of providing a more comprehensive representation of the dataset. In the revised manuscript, we have included representative images from all classes to better illustrate the variations of classes. This addition will enhance clarity and provide readers with a clearer understanding of the dataset's diversity. Additionally, we have improved the resolution of the images to ensure better visibility and interpretation. Kindly check section 3- sub section 3.1, 3.2.
  4. We have improved the resolution of Figures 4 to 16 to enhance clarity. Additionally, the confusion matrix in Figure 14 has been refined for better readability and visual presentation.
  5. We acknowledge the issue with Table 13 (Formerly it was Table 12 and updated the table no. to Table 13 as an additional Table (Table 2) has been added later) and have revised it to improve clarity and readability. The rows and columns have been properly aligned, and we have reformatted the table to ensure that the information is presented in a structured and comprehensible manner. We believe these modifications will enhance the ease of interpretation for the readers. We appreciate your suggestion and hope the revised version meets the required standards. Kindly check Section 4.
  6. We have now included a detailed comparison of the processing time for each individual model versus the proposed ensemble model. Table 9 highlights the analysis between accuracy improvements and processing time. The results demonstrate that the ensemble model achieves higher accuracy with less processing time. Kindly check Section 4.
  7. We have revised subsection "3.5. Optimization and Hyperparameter Tuning" to improve clarity and structure. The updated section now includes a more formal definition of the optimization process, along with the necessary mathematical formulations to enhance technical depth.
  8. We have now included the training loss and accuracy metrics for the proposed ensemble model to provide a comprehensive evaluation of its performance (Table 12). Additionally, we have incorporated comparative visualizations of loss and accuracy both with and without the Grasshopper Optimization Algorithm (GOA) (Figure 21). These graphs clearly demonstrate the impact of the optimization technique on convergence, overfitting reduction, and classification accuracy. Furthermore, we have added a detailed discussion on GOA that enhanced model generalization by optimizing hyperparameters such as learning rate, batch size, epochs, and dropout rate. The revised section now presents a clear comparison of training and validation accuracy as well as training and validation loss between the optimized and unoptimized models to justify the effectiveness of the optimization approach. Kindly Check Section 4.
  9. We have now included an analysis of the time efficiency of the Grasshopper Optimization Algorithm (GOA) in terms of training time for all the models. This addition helps to assess the computational trade-offs of using GOA, highlighting whether the optimization process significantly impacts efficiency while improving performance. The revised section provides a clearer understanding of GOA’s impact on model training duration and real-time applicability.
  10. The optimization process iteratively searches for the best hyperparameter values by minimizing the loss function and improving model convergence. By leveraging GOA, our model effectively balances exploration in the hyperparameter space, leading to improved generalization and enhanced classification performance. This discussion has been incorporated into the revised manuscript for better clarity. Kindly check Section 4.
  11. We have now included a brief discussion on the potential limitations of the proposed method, such as image overlapping and missing of primary data. Additionally, we have outlined future research directions, including exploring more efficient integrated and embedded frameworks for better clinical applicability. This discussion has been added to the revised manuscript to provide a more comprehensive perspective. Kindly check Section 5.

Reviewer 2 Report

Comments and Suggestions for Authors

In this study, a novel and optimized detection framework is proposed to classify Alzheimer's level. Three CNN models and one image transform model (ViT-B16) were trained with 0.234 million brain MRI images. The weighted average ensemble technique with Grasshopper optimization algorithm achieved 97.31% accuracy. The study contributes to the literature in terms of the research topic and the performance achieved. The paper clearly presents the proposed method and the necessity of the work. However, minor modifications are suggested.
- The reason for the choice of three of the CNN architectures is not explained. A comparison table would strengthen the rationale for choosing the selected architectures.
- The expressions in Figure 6 are not fully visible. Can be improved
- In the Conclusion section “The work offers promising performance including high accuracy of 97.31%, precision 97.32, recall 97.35 and recall 0.9732.” The recall value is given differently twice. It should be reviewed and corrected. 

Comments on the Quality of English Language

The quality of the English language is sufficient for me to understand the work.

Author Response

We sincerely appreciate for the time and effort to provide insightful and constructive feedback on our manuscript. Your valuable suggestions have significantly helped us refine and enhance the quality of our work. We have carefully addressed each comment and made the necessary revisions to improve the clarity, structure, and technical depth of our manuscript. Your guidance has been instrumental in strengthening our study, and we are grateful for your support in making this research more impactful. Thank you for your thoughtful evaluation and for helping us improve our work. All the corrections/modifications are highlighted in yellow marks on the revised manuscripts for your better understanding.

  1. We have now included a comparison table that highlights the key characteristics, strengths, and limitations of the selected CNN architectures. Additionally, we have elaborated on the rationale for choosing these architectures based on their performance, feature extraction capabilities, and relevance to the task of Alzheimer’s Disease diagnosis. This revision provides a clearer justification for our selection. Kindly check Section 3- Sub section 3.3.
  2. The quality of Figure 6 has been improved by adjusting the resolution and formatting to ensure that all expressions are fully visible and clear. This improvement will enhance readability and better convey the necessary information. Kindly check Section 3- Sub section 3.3.
  3. We carefully reviewed and corrected the inconsistency in the recall value to ensure accuracy and clarity in the Conclusion section. Kindly Check Section 5.

Thank you for your genuine and constructive feedback. We appreciate your assessment and have ensured clarity and coherence in our writing to maintain the quality of the English language throughout the manuscript.
